# Spatial Econometric Analysis of the Level and Influencing Factors of Coupling and Coordination between Regional Logistics and the Ecological Environment in China

**DOI:** 10.3390/ijerph192215082

**Published:** 2022-11-16

**Authors:** Xinbao Tian, Peiran Chen, Jie Li

**Affiliations:** School of Economics and Management, Shanxi University, Taiyuan 030006, China

**Keywords:** regional logistics, ecological environment, coupling and coordination relationship, entropy method

## Abstract

With the logistics industry becoming an important part of economic development, logistics activities have given rise to various environmental problems, which that affect the sustainable development of the ecological environment. In order to understand the status of regional sustainable development, it is of great significance to study the level of coordinated development of regional logistics and the ecological environment, and determine which variables will affect their coordinated development. To accomplish this objective, the entropy method is used to give the corresponding weight to each index and calculate the coupling degree and coordination degree of regional logistics and the ecological environment; next, the coupling coordination level of regional logistics and ecological environment in different provinces from 2010 to 2019 is measured by the coupling coordination degree model; then, we study the spatial effect of the influencing factors on the coordinated development level of regional logistics and ecological environment in 30 provinces of China from 2010 to 2019. At the same time, we investigate the spatial effect of each influencing factor on the coordination level. On the basis of the findings, relevant suggestions and measures are proposed. The main conclusions are as follows: (i) As China’s economy is in the process of transformation to high-quality development, the coordinated development of regional logistics and the ecological environment is still at a relatively low level; (ii) due to the differences in the development basis of each region, the coordinated development level of regional logistics and the ecological environment is characterized by a regional imbalance; (iii) because of the significant spatial spillover effect, the coordinated development level of regional logistics and ecological environment shows obvious correlation in space.

## 1. Introduction

With the rapid economic development of China, environmental problems such as smog and sandstorms have followed, which are due to the negative impact of economic development on the environment. The regional ecological environment is a complex systemic whole, that is affected by a variety of factors. Logistics, as a bridge connecting the various systems of social development, has an increasing impact and role in various industries, especially in industrial transformation and upgrading, reducing costs, and improving efficiency [1]. Furthermore, the logistics of packaging, distribution processing, warehousing, transportation, and distribution of logistics activities, has a certain impact on the ecological environment, as shown in Figure 1.

Because the functions of the systems are mutual, the ecological environment will also play the following two roles in logistics activities: (i) Promoting the development of green logistics. A good ecological environment can bring economic benefits as well as ecological benefits, which will improve people’s awareness of environmental protection. Under the background of the concept of green development and the promotion of environmental protection, the green development of the logistics industry will be promoted, which is guided by national policies, such as developing new energy transportation tools and packaging materials, and strengthening supervision of pollution behaviors. (ii) Increasing the cost of logistics transformation caused by environmental deterioration. Environmental deterioration will lead to the government’s efforts to manage the environment, and new industries emerge as the times require, such as the sharing economy and new energy industry, which will further promote the transformation and upgrading of the logistics industry, while the use of high-energy-consuming transportation tools and non-degradable packaging materials will be restricted. The processes involved in the effect of ecological environment on logistics is shown in Figure 2.

It can be seen that regional logistics and the ecological environment interact and restrict each other. The interaction mechanism of the two systems closely follows the law of interaction and coupling. As an important part of the regional economy, the matching degree of the two systems of regional logistics and ecological environment can largely reflect the coordination level of economic development and green development. Therefore, a deep understanding of the coupling law is of great significance for regional sustainable development to grasp the characteristics of their coordinated evolution. Based on this, in this study, we measure the regional logistics and ecological environment coupling coordination, carry out a deep dynamic analysis of its spatio-temporal evolution characteristics, and introduce a spatial measurement model, which is a relatively advanced and mature method for spatial research on inter-regional elements. It is particularly effective to deal with spatial interaction (spatial autocorrelation) and spatial structure (spatial heterogeneity/spatial heterogeneity) in the regression model using cross-sectional data and panel data. In this paper, the spatial econometric model will be used to test the spatial autocorrelation, and the main factors affecting the coordinated development will be analyzed. The findings will help the government re-examine the relationship between regional logistics development and ecological environment to further promote green development and provide a basis for the direction of policy measures.

Current research is mainly focused on the following areas.

(i) Research on the impact of regional logistics on the environment. With the rapid development of the logistics industry, a series of problems have emerged, the most serious of which is increasing environmental pollution. Because logistics mainly depends on various means of transportation, pollution is inevitably created during the transportation process. This pollution is mainly caused by pollutants such as carbides and sulfides produced by the combustion of fossil energy, which mainly pollutes the atmospheric environment. Nowadays, many scholars pay more attention to the topic of carbon emissions in logistics activities. Konur et al. pointed out that the logistics industry has caused a series of environmental problems in the development process, such as energy consumption, carbon emissions and air pollution [2]. Yang et al. believed that developing low-carbon logistics and changing the development mode of the logistics industry were the best ways to build a low-carbon economy, which is related to whether the overall economic environment can achieve low-carbon emissions [3]. Liddle showed that the transport sector was important in economic growth, which was not conducive to environmental quality [4]. At present, the increase in environmental pollution caused by transportation is an important determinant of whether developing countries can achieve sustainable development. Studies by many scholars have shown that emission reduction policies and logistics decisions are important factors affecting carbon emissions and the ecological environment; for example, Papagiannaki et al. discussed the influencing factors of carbon emissions of the logistics industry [5], Daryanto studied emission reduction policies [6], Kaur et al. studied logistics decisions [7], Behnke et al. studied vehicle routes [8], and Choi studied carbon emission taxes [9]. At the same time, many scholars have also studied the factors affecting logistics carbon emissions. For instance, Geng et al. discussed the factors affecting the carbon emissions of the logistics industry, and then put forward solutions and suggestions in order to promote the low-carbon development of the logistics industry [10]. Bao et al. analyzed the impact of five factors, including energy structure, on the carbon emission scale of regional logistics by quantitative analysis [11]. Zhao et al. studied the energy consumption of the logistics industry in Beijing, Tianjin, and other regions, quantitatively calculated the carbon emissions generated by regional logistics activities, and proposed solutions for the development of the region. It strongly supported the coordinated development of the region [12]. Dong et al. used the SBM-Undesirable and Super-SBM-Undesirable model to measure the ecological efficiency of the logistics industry and the three types of improvement potential, and the ESDA method was used to analyze and explore the typical regions and linkage regions for the development of pollution reduction potential at the spatial evolution level [13]. Yang et al. focused on the impact of urban logistics on the environment, analyzed the internal relationship between the environmental impact of logistics facilities and land use, and further revealed the important impact of the spatial evolution of logistics facilities on urban layout [14].

(ii) Research on the coordination of economic development and ecological environment. Many scholars have performed some research on the coordination of economic development and the environment using the coordination degree model. For example, Wang et al. used a combination of quantitative models to calculate and analyze the coordinated development of economic development and environment in several regions; the results showed that the correlation and coordination degree between subsystems were weak, and the performance of coordination degree level in several regions was less satisfactory [15]. Zhang et al. used a combination of correlation and coordination models to study the coordination between economic development and the environment in several regions [16]. Zhang used a dynamic coupling model to analyze the changes in the level of coordination in order to find the relevant laws of change and provide support for related research, which showed that the current coordination needs to be further strengthened [17]. Chen et al. used a coordination model to study the changes in the level of coordination between the industrial economy and ecological environment [18]. Feng et al. regarded the internal mechanism of ecological service-oriented economy as the core, and adopted the Sanjiangyuan Nature Reserve in Qinghai Province as a typical case study sample to make some innovations in the coordination mechanism between economic development and ecological environment in China’s deeply impoverished areas [19]. Zhang et al. built a coupling coordination degree model and social network analysis by building an evaluation index system for ecological protection and high-quality economic development, where the relevant data of 75 prefecture-level cities in the Yellow River basin was used to explore their coordination and linkage effects and network structure characteristics [20]. It can be seen that studies mostly focus on the economic development and the ecological environment, most of which adopt the coordination model. This paper holds that there is also a close relationship between logistics and the ecological environment. The reason is that the modern logistics industry is a composite service industry integrating transportation, warehousing, information, and other areas, covering almost all social production and consumption fields. It plays an important role in organizing and promoting resource allocation and industrial upgrading of various industries, and is a strong pillar and artery for the development of the national economy. The ecological environment is also affected by the development of various industries to a large extent; thus, there must be an interaction mechanism between them. Therefore, this paper refers to the practice of predecessors and switches the perspective of research that is, from the perspective of logistics rather than the economy, the coordination degree model used to study the coordination relationship between logistics and ecology, where the spatial factor is added to further analyze its spatial distribution.

(iii) Research on the extended application of coupling coordination theory. Zhou et al. believed that “coupling” is a phenomenon originating from physical science, which can be used to observe and measure the relationship between economic phenomena and the impact of systems on each other [21]. For example, Srinivasan et al. investigated the relationship between urbanization and water resources [22]. Fan et al. evaluated the trend in coordinated development of urban social economy and ecological environment [23]. Li et al. analyzed the relationship between urbanization and ecological environment [24]. Chen et al. discussed the coordination between social economy and carbon emissions [25]. Song et al. focused on the relationship between carbon emissions and urbanization [26]. Kang et al. focused on the coordination of ecological protection and economic development of 46 cities along the Yellow River basin, exploring the spatio-temporal evolution and influencing factors of ecological environment protection and high-quality development of cities by using the coupling coordination model and geographical detectors [27]. Tang et al. performed quantitative analysis of the economic development and ecological environment of Shaanxi Province from 2008 to 2018, analyzing the coupling and coordination level and evolution characteristics of the two systems as well as the main influencing factors behind them [28].

The coupling theory has been widely studied in many fields, and there has also been some progress in studies on the coupling of logistics carbon emissions and the ecological environment. However, relatively few studies take spatial factors into account. This paper introduces a spatial econometric model to conduct a dynamic analysis of the spatial evolution characteristics of the coordinated development level; selects influencing factors to empirically analyze how each factor affects the coupled coordinated development level in space; and analyzes the spatial effect of each factor on the coupled coordinated development level of logistics and the ecological environment. In summary, the paper makes an empirical contribution to literature on the coupling and coordinated development of regional logistics and ecological environment by using data from China’s provinces from 2010 to 2019. The research conclusion further confirms that the coupling and coordination of regional logistics and the ecological environment are significantly affected by the spatial spillover effect, that is, improvement in the coupling and coordination level of adjacent areas has a positive spillover effect on the coupling and coordination level of the region, which will have an enhancing effect on regional development.

## 2. Research Methods

### 2.1. Construction of the Index System

The index system was established to accurately represent the indicators of the logistics subsystem and the ecological environment subsystem, adopting the method of combining qualitative and quantitative research. Twenty-one secondary indicators are established in this paper as shown in Table 1; among them, twelve second-level indicators represent the regional logistics level. In terms of the selection of indicators for the regional logistics development level, this paper refers to the index design of Qin for the efficiency of regional logistics industry [29], which is based on the principles of representativeness, feasibility, and pertinence, and considers differences in research objects and making reasonable innovation. Among them, the dimension of development scale represents the size of the regional logistics industry development, which is mainly evaluated from the quantitative aspects, including the number of enterprises, output value, business volume and other quantitative performance indicators. The dimension of development level focuses on the development quality, including four indicators, namely, the number of employees in the logistics industry, the average wage of employees, the average output value, and fixed asset investment. The dimension of informatization level selects mobile phone penetration rate as the measurement indicator to analyze and control logistics information. The dimension of development potential chooses the incremental ratio of industrial development and employment growth for measurement. In terms of the selection of indicators for the ecological environment, this paper establishes nine secondary indicators representing the level of ecological environment development by referring to the idea of taking resource conservation and environmental protection as measures of green development in Guo’s research on the assessment indicators of regional high-quality development mode [30]. Among them, the dimension of environmental pollution selects carbon dioxide emissions, nitrogen oxide emissions, and particulate emissions as environmental pollution indicators. The dimension of environmental remediation selects the domestic waste clearing and transportation volume and the number of harmless treatment plants as the measurement indicators. In the dimension of natural resources, water resources, as the core resource of the ecosystem, are a necessary condition to ensure the ecological balance and development, so the indicator of total water resources is selected. Finally, in the dimension of development potential, this paper adopts the relative incremental measurement index, and selects the growth rate of industrial pollution control, the growth rate of afforestation area, and the growth rate of ecological restoration investment.

Considering the extensive consumption of fossil energy in the logistics industry and the increasingly serious destruction of the ecological environment, this paper takes provinces as the research objects. According to the availability, data are collected during the last ten years for statistical analysis and used to explore the changes in the coordination level of regional logistics and environment in various provinces. The data are mainly obtained from the *China Statistical Yearbook* and *China Energy Statistical Yearbook*. The data in the statistical yearbooks of various provinces and the data in the statistical reports are taken as important supplements. Thirty provinces are selected as research samples because of the scarcity of various indicators in individual provinces.

### 2.2. Determination of Indicator Weights

To determine the weights of the indicators, this paper uses the entropy method, which is a method for calculating and allocating index weights to the indicators according to objectively collected values [29]. Therefore, the results obtained using this method are relatively objective. Because the dimensions and signs of each index are different, direct calculation will cause large errors. This paper uses the range method to standardize the data. Because positive indicators represent the positive impact on the system, and negative indicators represent the negative impact on the system, in order to avoid inaccuracy of the results caused by the different directions of impact of the indicators, the processing methods are slightly different. The formula is as follows:(1)$$y_{ij}=\frac{x_{ij}-\min\left(x_{ij}\right)}{\max\left(x_{ij}\right)-\min\left(x_{ij}\right)}\;(\mathrm{Positive})$$
(2)$$y_{ij}=\frac{\max\left(x_{ij}\right)-x_{ij}}{\max\left(x_{ij}\right)-\min\left(x_{ij}\right)}\;(\mathrm{Negative})$$
where *Y_ij_* is a standardized data matrix; *X_tj_* is a matrix unit of *i* rows and *j* columns of original data; *i* represents provinces; and *j* represents the *j*th index measured in the article; *i* representing the province and *j* representing the *j*th indicator measured in the article.

The proportion of province *i* under the *j*th indicator is as follows:(3)$$q_{ij}=\frac{y_{ij}}{\sum_{i=1}^{m}y_{ij}}$$

Entropy measurement of the *j*th index is as follows:(4)$$e_{j}=-k\sum_{i=1}^{m}q_{ij}\ln\left(q_{ij}\right)$$
where *k* = 1/ln(*m*), and *k* > 0, *e_j_* ≥ 0.

Redundancy measure is as follows:(5)$$d_{ij}=1-e_{ij}$$

Index empowerment is as follows:(6)$$\lambda_{ij}=\frac{d_{ij}}{\sum_{j=1}^{n}d_{ij}}$$

The weights of 21 secondary indicators representing the development level of regional logistics and ecological environment are calculated by entropy method, as shown in Table 2 and Table 3.

### 2.3. Measurement of the Level of Coupling Coordination

The coupling coordination model can measure the development and coordination level between different systems. In order to scientifically and comprehensively measure the coordination degree between regional logistics and the ecological environment, this paper builds a coupling coordination degree model by referring to Tang et al., Jin et al., and Ge et al. [31,32,33]. If *a* and *b* represent the regional logistics system and the ecological environment system, respectively, *Y_aj_* and *Y_bj_* are the values of the *j*th index of the regional logistics system and the ecological environment system, respectively, and the value of *j* is *j* = 1, 2, 3, …, *N*, which are standardized by using the Formulas (1) and (2).

This paper uses the research methods of Tang et al. for reference and uses entropy method to determine the weight [34], by which the development level of regional logistics and ecological environment system can be obtained, respectively. The system level score calculation formula is as follows:(7)$$Z_{i}=\sum_{j=1}^{n}\lambda_{ij}y_{ij}\;\cdot \;\sum_{j=1}^{n}\lambda_{ij}=1$$
where *n* represents the number of indicators, and $\lambda_{aj},\lambda_{bj}$ are the weight of the *j*th indicator of the two systems. *Z_i_* is the comprehensive level of regional logistics subsystem and ecological environment subsystem.

The coupling degree model between different systems is shown in Formula (8):(8)$$C\left(Z_{1},Z_{2},\ldots ,Z_{L}\right)=L*{\left[\frac{Z_{1}Z_{2}\ldots Z_{L}}{{\left(Z_{1}+Z_{2}+\ldots Z_{L}\right)}^{L}}\right]}^{\frac{1}{L}}$$
where *L* = 1, 2,… represents the number of systems. When *L* = 2, the formula is as follows:(9)$$C_{ab}=2*{\left[\frac{Z_{a}Z_{b}}{{\left(Z_{a}+Z_{b}\right)}^{2}}\right]}^{\frac{1}{2}}=2\frac{\sqrt{Z_{a}Z_{b}}}{\left(Z_{a}+Z_{b}\right)}$$
where *C_ab_* represents the coupling degree between the regional logistics system and the ecological environment system, with a value range of 0–1.

In order to avoid the inaccurate results of the regional logistics system *Z_a_* and the ecological environment system *Z_b_* when the values are small at the same time, the paper establishes a coupling coordination degree model that can obtain reasonable results. The formula is as follows:(10)$$D_{ab}={\left(C_{ab}*T_{ab}\right)}^{\frac{1}{2}}$$
(11)$$T_{ab}=\alpha Z_{a}+\beta Z_{b}$$

*D_ab_* represents the coupling coordination degree of the regional logistics system and the ecological environment system, with a value range of 0–1. *T_ab_* is the comprehensive evaluation index of the regional logistics system and the ecological environment system. *α*, *β* are undetermined coefficients, respectively, whose value is *α* = *β* = 0.5.

According to Zhan et al. and in combination with the actual situation, the definition criteria for the degree of interaction coupling are shown in Table 4.

When the value of coupling coordination degree is smaller, the level of coupling coordination is lower. With a gradual increase in the value, the coordination level changes from coupling imbalance to coupling excess, and finally reaches the stage of high-quality coupling coordination. Regional logistics and ecological environment are at a low level, which indicates that due to their unbalanced development, there is no beneficial interaction between regional logistics and the ecological environment, and the effect of mutual promotion and development cannot be achieved. On the contrary, they will achieve a common progress and promotion. In terms of the development stage of regional logistics and ecological environment, when $Z_{a}^{n}>Z_{b}^{n}\;$, it indicates that the regional logistics comprehensive index of the *i*th province unit or region in the *n*th year is greater than that of the ecological environment comprehensive index. At this time, the development level of the ecological environment is lagging; that is, the promotion effect of logistics on the ecological environment is greater than that of the ecological environment on the development of logistics. However, when $Z_{a}^{n}<Z_{b}^{n}\;$, the development level of regional logistics is relatively low compared with that of the ecological environment, and the promotion effect of logistics on the ecological environment is weaker than that of the ecological environment on logistics. At this time, the green development of logistics should be accelerated to promote development of the ecological environment so as to achieve a beneficial coupling stage.

## 3. Measurement of the Level of Coupled and Coordinated Development between Regional Logistics and Ecological Environment

### 3.1. Overall Coordinated Development Level and Characteristics

According to the calculated regional logistics and ecological environment values, the coupling coordination level is calculated and the results are shown in Table 5. From the perspective of national average, the level of coupling and coordination between regional logistics and the ecological environment remains between 0.52 and 0.60 from 2010 to 2019, which indicates a very low level of coordination. From the perspective of the provincial level, the development state increases first and then declines.

In terms of the ratio between the coefficient of regional logistics comprehensive development level and the coefficient of ecological environment comprehensive development level, *Z_a_/Z_b_* was greater than 1 in 2013, which indicated that the development level of the ecological environment was in a lagging stage. In 2018, *Z_a_/Z_b_* was less than 1, which indicated that the environmental development level was improved, while the development level of regional logistics was at a relatively backward stage. Around 2018, due to the impact of Sino-US trade friction, China’s manufacturing industry was severely impacted, which also impacted the logistics industry and caused the development of logistics to lag behind that of the ecological environment.

It can be seen that the maximum value of coupling coordination degree is all in Guangdong from the data of provinces during all the years, it basically keeps around 0.8, and basically reaches the stage of good coordination. However, at the same time, it should also be noted that there are some regions with a low level of coordinated development which mainly included Gansu, Qinghai and Ningxia. The main reason for the low degree of coupling coordination of these provinces is the low level of logistics industry development and the high level of ecological environment development, which leads to the low level of coordination between them.

### 3.2. Level and Characteristics of Coordinated Development in Different Regions

In this paper, China is divided into seven regions to investigate the coordinated development level of different regions. The results can be seen in Table 6 and Figure 3. The coupling and coordination levels of all regions are relatively stable except for the sharp decline in 2013 and 2019. Among them, the coupling coordination level of Central China, East China, and South China is the highest among the seven regions, and the average coordination level has reached the primary coordination level. North China, Northeast China, and Southwest China are at the barely coordinated level, while Northwest China is at the verge of imbalance. The overall coordinated development level is slowly rising in fluctuations.

## 4. Analysis of the Factors Affecting the Coordinated Development of Regional Logistics and Ecological Environment

### 4.1. Determination of Influencing Factors

Coordinated development between regional logistics and ecological environment is the result of the comprehensive action of various factors, which is not only driven by internal factors such as the level of scientific and technological development, the level of economic development, and the industrial structure, but also by external factors such as the degree of industrial pollution control, the level of transportation facilities, and the degree of garbage control. At the same time, because regional logistics and ecological environment have spatial spillover effects, spatial factors should be taken into account in the development of different regions. The following six influencing factors are selected:(i)The level of scientific and technological development (*X*_1_). Science and technology can promote high-level development of the logistics industry. Scientific and technological innovation can effectively drive the sustainable development of the logistics industry and promote the organic combination of human, financial, and material resources of the logistics industry. Science and technology are also of great significance to the improvement in the ecological environment, especially in environmental protection and green production. Technological innovation will play a catalytic role in promoting the development level of regional logistics and improving the ecological environment.(ii)The level of economic development (*X*_2_). The level of regional economic development is closely related to the development of the logistics industry. On the one hand, economic development will bring huge logistics demand and promote the expansion of the logistics industry. On the other hand, the improvement in economic level will stimulate the progress of production technology and optimize the overall input-output efficiency of the logistics industry. In addition, high-quality economic development contributes to the improvement in the ecological environment.(iii)The degree of industrial pollution control (*X*_3_). The degree of industrial pollution control reflects the level of security provided by a region for sustainable development.(iv)The industrial structure (*X*_4_). Due to the high correlation between the logistics industry and the industrial structure, the adjustment and upgrading of the industrial structure can generate demand for logistics services and bring opportunities for the development of the logistics industry. At the same time, a reasonable industrial structure will help reduce the pressure on the ecological environment and lower the cost of environmental governance.(v)Level of transport facilities (*X*_5_). A complete transportation infrastructure is the basis for the stable development of the logistics industry, which will make the distribution of the regional logistics warehousing base more convenient, save transportation costs on a large scale, accelerate the efficiency of resource circulation, optimize resource allocation, and improve the development level of regional logistics.(vi)Waste management degree (*X*_6_). The degree of waste management is an important indicator to reflect the regional ecological environment. Coupling coordination degree level (*Y*) is used as the explained variable, as shown in Table 7. In addition, descriptive statistics of variables are presented, as shown in Table 8.

### 4.2. Measurement of Spatial Correlation

A spatial correlation test is the first step of spatial econometric analysis. Only data with spatial correlation can be used for spatial econometric analysis. This paper tests the spatial correlation of the regional logistics comprehensive index, the ecological environment comprehensive index, and the coupling coordination comprehensive index, and obtains the Moran’s I and *p* value data, as shown in Table 9. On the whole, the values of the three indicators are not zero, which indicates that the data are spatially related. The comprehensive index of ecological environment in 2011, 2013, and 2014 did not pass the significance test, with a *p* value of less than 0.05. As for the coupling coordination degree, except in 2014, all other indicators passed the significance test with a *p* value less than 0.05. This shows that the data have spatial correlation and can be used for quantitative analysis. It can be seen from Table 9 that the minimum value of the coordinated comprehensive index of regional logistics and ecological environment is 0.11 and the maximum value is 0.34, which indicates that the coupled coordinated development level of regional logistics and ecological environment has a positive correlation in space.

### 4.3. Selection of Econometric Models

The above autocorrelation test results show that most of the data pass the spatial autocorrelation test, which means that the impact of spatial factors should be considered in the study of the coupling coordination level of regional logistics and the ecological environment.

Selection of econometric models. The LM Test is used to verify whether SEM or SLM is more suitable for this study. It can be seen from the results in Table 10 that the Lagrange multiplier index test results of the spatial error and spatial lag models are 23.01 and 39.466, respectively, and they are also statistically significant at the 1% level. Moreover, it can be seen that the results of R-LM lag do not pass the test; thus, we reject the original hypothesis. Therefore, preliminary judgment indicates that the SLM model should be selected.

(i) Model test. The results of Wald and LR test models are used to determine whether the spatial Dobbin model will degrade. SDM is used for lag analysis under the condition that there is no degradation. Otherwise, if there is degradation, spatial error or spatial lag models may be used for subsequent research. It can be seen from the results in Table 10 that both of them have passed the test; it shows that the spatial Dobbin model cannot be degraded, so the spatial Dobbin model should be selected for the subsequent spatial metrology research in this paper.

(ii) Determination of model. The use of fixed models was determined by Hausmann test.

### 4.4. Spatial Effect Analysis

The estimation results in Table 11 show that the likelihood function value of the spatiotemporal double fixed model reaches 648.20, whereas the other two models achieve smaller values. Therefore, this paper uses this model in the empirical analysis of subsequent econometric studies. The estimation results of the three estimation models in Table 11 show that although the regression coefficient of the time fixed model is not statistically significant, the values of the spatial autoregressive coefficient rho of the regional logistics and ecological environment coupling coordination level for the other two models are 8.63 and 1.20, and are significant at the 1% level and 10% level, respectively. The results show that the coordinated development level of adjacent areas will have a positive spillover on the local area, which improves the coupled coordination level of regional logistics and the ecological environment.

The regression coefficient of scientific and technological development on the coordinated development level was 1.76, which passed the test at the level of *p* value 0.1, and the lag term was 2.87, which passed the test at the level of *p* value 0.01. This showed that the level of scientific and technological development promotes rather than suppresses the level of coupled and coordinated development, and the surrounding areas also promote the level of coordinated development. As an important power source to drive the green development of logistics, the level of scientific and technological development can promote the coordinated development of logistics and the ecological environment. At the same time, the scientific and technological development of the surrounding areas has significantly promoted the local coordination level, which indicates that the scientific and technological exchanges among regions have promoted the coordination level.

The regression coefficient of economic development on the coordination level was −0.71, it failed the test. The regression coefficient of lag term was 2.10, which passed the test with a *p* value of 0.05. This showed that the economic development of the neighboring areas had a positive impact on the coordinated development of local regional logistics and the ecological environment. On the one hand, it may be that the development of the ecological environment has not been paid enough attention in the process of economic development in the region; it has not significantly affected the level of coordinated development. On the other hand, as an external variable, the economic development of neighboring regions will have corresponding impacts on the ecological environment and logistics development due to the operation of spillover effects, and will further affect the coordinated development level of the region.

The regression coefficient of the degree of industrial pollution control on the coordination level was −0.33; it failed the test. The regression coefficient of the lag term is −1.89; it passed the test at the level of *p* value 0.05. This showed that the industrial pollution control in this area had a restraining effect on the coordinated development level, although it was not significant. The industrial pollution control in the surrounding areas had significantly inhibited the coordination level of the region.

The regression coefficient of industrial structure change was the largest at 1.90; it passed the test at the level of *p* value of 0.1, and had a positive effect on coordinated development. The regression result of lag term was 0.11; it failed the test. This that the adjustment of the industrial structure in this region had significantly promoted shows the development level of the coordination degree, while the adjustment of the industrial structure in the adjacent regions had not played a significant role.

The regression coefficient of transportation facilities on the coordination level was 1.08; it failed the test. The regression result of the lag item was 0.71. The test results showed that the impact of the development level of transportation facilities was not obvious in the region or in the adjacent regions.

The regression coefficient of waste treatment degree on the coordination level was 1.86 and is statistically significant. The regression result for the lag term was −3.07 and was significant at the 1% level. The results showed that the local ecological environment level could be significantly improved through garbage treatment, which indicated that the level of garbage treatment was an important factor in determining whether the two systems were coordinated. The surrounding areas had a significant inhibitory effect on the local area, probably because the garbage treatment level of the adjacent areas will play a certain exemplary role in the local area, thus affecting the coordinated development level of the local area.

### 4.5. Decomposition of the Spillover Effect

Because of the lag term in the spatial Dobbin model, the measured regression coefficient cannot accurately reflect the effect of each influencing factor. In order to solve this problem, the spatial effect of these factors is further divided into direct effect, indirect effect, and total effect for further analysis. The results can be seen in Table 12.

In terms of direct effects, the level of economic development (*X*_2_), the level of industrial structure (*X*_4_), and the degree of garbage treatment (*X*_6_) have a significant positive effect on the level of coupling coordination. The elastic coefficients are 4.30, 4.87, and 9.29, respectively, and all are significant at the 1% level. Among them, the elasticity coefficient of garbage treatment degree is 9.29, which indicates that at present, the degree of garbage treatment is one of the important factors determining whether regional logistics and ecological environment can develop harmoniously. Because the improvement in garbage treatment level has greatly improved the environmental level, the coupling coordination level has been improved. In addition, the improvement in the level of industrial structure has promoted the transformation of the secondary industry into the tertiary industry, which reduces the pollution of the secondary industry to the environment and plays a positive role in protecting the ecological environment, resulting in the improvement in the level of coupling coordination. Economic development also shows a positive and positive impact on whether the two systems can develop harmoniously and the degree of coordinated development.

In terms of indirect effects, the level of scientific and technological development (*X*_1_) has a significant positive effect on the level of coupling coordination, while other influencing factors have an inhibitory effect on the level of coupling coordination. Among them, the improvement in the industrial structure level in the adjacent areas has no significant inhibitory effect on the level of coupling coordination in the region, while other indicators have a significant inhibitory effect. The improvement in garbage treatment degree has the most obvious inhibitory effect on this area, and the elastic coefficient is −3.83, and is significant at the 1% level. It may be that with the improvement in economic level, there is a surplus of garbage but the treatment facilities cannot meet the development needs, resulting in a low harmless treatment rate of garbage. At the same time, the impact of garbage disposal on the environment also has an indirect negative impact on the environmental level of the adjacent areas.

In terms of the total effect, industrial pollution control and transportation facilities have a significant inhibition effect on the level of coupling coordination degree, of which the elastic coefficient of industrial pollution control on the level of coupling coordination degree is −5.37 and the degree of inhibition is the largest. Whenever the level of industrial pollution control increases by 1%, the level of coordination degree decreases by 5.37%. This shows that improving the level of environmental governance will greatly change the level of coupled and coordinated development of the ecological environment and regional logistics. Therefore, industrial pollution control and transportation facilities should be improved simultaneously to maintain the level of coordinated development within a reasonable range.

### 4.6. Robustness Test

In order to ensure reliability of the model estimation results and avoid the effect of endogenous variables, the weight matrix is reselected to build the model and estimate and compared with the original weight matrix estimation results. The results are shown in Table 13. The direction of the effect of each influencing factor on the coupling coordination level is consistent, and there is no significant difference. Therefore, the results obtained by using the original model accurately reflect the level of coupling and coordinated development of various influencing factors on regional logistics and the ecological environment, and the results can be trusted.

## 5. Conclusions

This paper uses the entropy weight method to evaluate the development level of regional logistics and the ecological environment in 30 provinces across China. On this basis, their coupling coordination level is measured, and the six main factors affecting the coordination level are analyzed using the spatial econometric model. Based on empirical research, three conclusions are drawn as follows:

(i) The coordinated development of regional logistics and the ecological environment is still at a relatively low level. From the perspective of overall coordination level, the coupling coordination level of regional logistics and the ecological environment was in the transition from the reluctant coordination type to the primary coordination type from 2010 to 2019. The comprehensive development level of the logistics subsystem is relatively stable, while the ecological environment system shows the phenomenon of developing in fluctuations. The ratio of the comprehensive index of the logistics subsystem to the comprehensive index of the ecological environment subsystem shows a fluctuating upward trend. Each province, as well as the whole country, has shown a development situation of first rising in fluctuations and then falling.

(ii) The coordinated development level of regional logistics and the ecological environment is characterized by regional imbalance. From the regional perspective, the coupling coordination degree of Central China, East China, and South China is the highest among the seven regions, and the average coordination level has reached the primary coordination level. North China, Northeast China, and Southwest China are at the barely coordinated level, while Northwest China is at the verge of imbalance, where the overall coordinated development level is slowly rising in fluctuations.

(iii) The coordinated development level of regional logistics and the ecological environment shows obvious correlation in space. This paper tests the spatial correlation of the regional logistics comprehensive index, the ecological environment comprehensive index, and the coupling coordination comprehensive index, and obtains the corresponding data. On the whole, the level of coupled and coordinated development of regional logistics and ecological environment has a positive correlation in space; in other words, the improvement in the level of coupled and coordinated development in adjacent areas has a positive spillover effect in this area.

Our research also has some limitations. First and most notably, our results are based on data for China. We believe that the results are correct based on the empirical test above and conform to China’s national condition. However, they are not widely applicable to other regions because of the different situation of each region. Therefore, we can expand the research scope by comparing relevant data from other countries and study the level of coupling and coordination relationship between regional logistics and ecological environment. At the same time, in further studies, the research perspective can be expanded, and the research object can be extended to prefecture-level cities to study the development of urban coupling and coordination.

Based on the above conclusions, the following suggestions are put forward for the development of regional logistics and the ecological environment:

First is strengthening interregional cooperation and exchanges. Because of the unbalanced development between regions, it is necessary to further strengthen interregional cooperation and exchanges. At present, the development level of the eastern coastal areas is generally high, industrial transformation and upgrading are continuing, and most of the regional economies have developed from relying on the secondary industry to relying on the tertiary industry. Most of the central regions are in the period of industrial transformation and upgrading and are optimizing their industries, mainly relying on the secondary industry to drive the local economy. However, the level of economic development in the western region is relatively low, there are only a small number of regions relying on the secondary industry, and most of them rely on the primary industry. Therefore, interregional cooperation and exchange should focus on industrial transfer, adjusting the industrial structure, driving the development of underdeveloped areas, and promoting the balanced and coordinated development of regions.

Second is determining the development direction based on regional characteristics. The coupled and coordinated development level of logistics and the ecological environment in the central and western regions is relatively low. In the process of economic development, we should try our best to reduce pollution emissions to protect the ecological environment, and actively implement the concept of green development. The logistics development level in the eastern region is high; however, the pollution to the ecological environment is low, which causes a high degree of coupling coordination. The logistics development level in the central region is low, and there is extensive damage to the environment in the process of development. The logistics development level in the western region is low, but the ecological environment is good, which causes the central and western regions to show a low level of coordinated development. The eastern region should continue to maintain the current development trend, while the central and western regions should accelerate industrial transformation and upgrading, optimize industrial structure, and pay attention to environmental protection. The development of the economy and logistics industry has brought a lot of harm to the environment; in particular, the rapid development of the secondary industry has caused damage to the ecological environment, causing regional logistics to continue to develop, but the ecological environment to be negatively affected. Therefore, we should pay attention to the parallel development of environmental protection while developing the economy.

Third is optimizing industrial structure and promoting industrial transformation and upgrading. In the process of development, each region should upgrade its existing industries or adjust its industrial structure according to regional characteristics, which would effectively promote the coordinated development of regional logistics and the ecological environment. With the continuous optimization of the industrial structure, and transformation and upgrading of the secondary industry to the tertiary industry in the eastern and central regions, the coupled and coordinated development level of regional logistics and the ecological environment will continue to improve and move towards a good coupled and coordinated stage. In the process of industrial transformation and upgrading, the western region should pay attention to the protection of the ecological environment so as to improve the level of coupling and coordination between regional logistics and the ecological environment while developing the economy.

## Figures and Tables

**Figure 1 ijerph-19-15082-f001:**
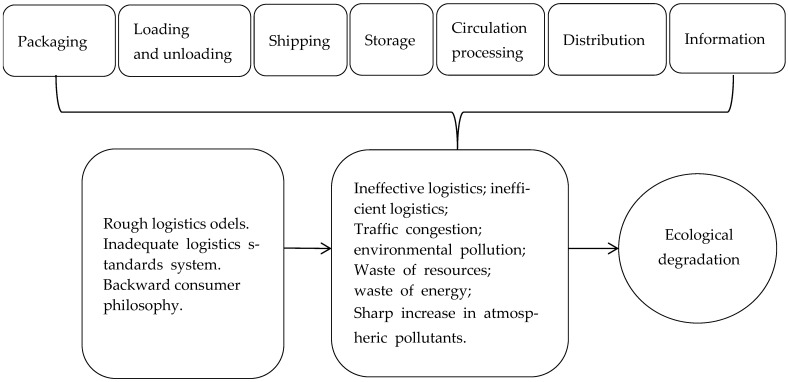
Mechanisms of action by which logistics activities exert an impact on the ecological environment.

**Figure 2 ijerph-19-15082-f002:**
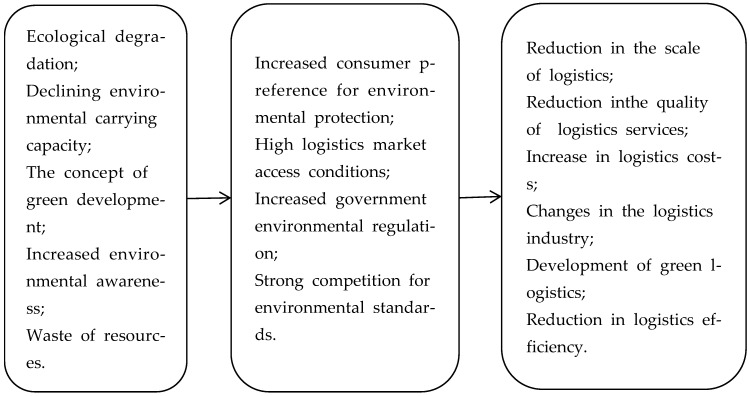
Mechanism underlying the role of the ecological environment in logistics activities.

**Figure 3 ijerph-19-15082-f003:**
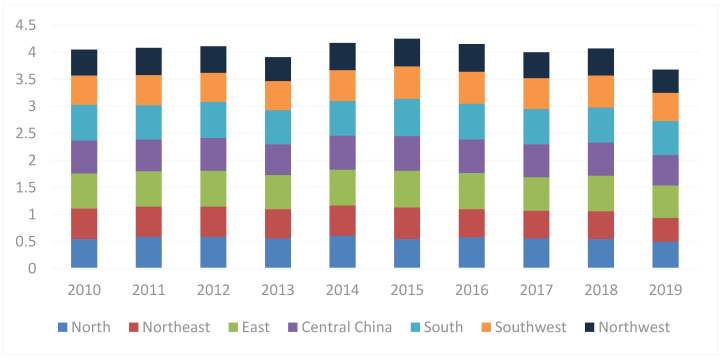
Changing trend in coupling coordination level from 2010 to 2019.

**Table 1 ijerph-19-15082-t001:** Regional Logistics and Ecosystem Indicator System.

Subsystems	First-Level Indicators	Variables	Second-Level Indicators	Indicator Attribute
Regional logistics subsystem (X)	Scale of development	X(11)	Number of legal person units of logistics enterprises (pcs)	Positive
		X(12)	Regional logistics output (CNY billions)	Positive
		X(13)	Cargo volume (million tonnes)	Positive
		X(14)	Cargo turnover (billion tonne kilometres)	Positive
		X(15)	Post and telecommunications business volume (CNY billions)	Positive
	Level of development	X(21)	Employment in logistics (10,000)	Positive
		X(22)	Average wage of employed persons (CNY)	Positive
		X(23)	Average output value of employed persons (CNY millions)	Positive
		X(24)	Investment in fixed assets (CNY billions)	Positive
	Level of informatization	X(31)	Mobile phone penetration rate (units per 100 population)	Positive
	Development potential	X(41)	Growth rate of output (%)	Positive
		X(42)	Employment growth rate (%)	Positive
Ecosystem subsystem (Y)	Environmental pollution indicators	Y(11)	Sulphur dioxide emissions (million tonnes)	Negative
		Y(12)	Nitrogen oxide emissions (million tonnes)	Negative
		Y(13)	Particulate matter emissions (million tonnes)	Negative
	Environmental remediation indicators	Y(21)	Volume of domestic waste removed (million tonnes)	Positive
		Y(22)	No. of environmentally sound treatment plants (no.)	Positive
	Natural resource indicators	Y(31)	Total water resources (billion cubic metres)	Positive
	Development potential indicators	Y(41)	Growth rate of industrial pollution treatment volume (%)	Positive
		Y(42)	Growth rate of afforested area (%)	Positive
		Y(43)	Growth rate of investment in ecological restoration (%)	Positive

**Table 2 ijerph-19-15082-t002:** Indicator weights for regional logistics.

Indicators	Number of Legal Entities in Logistics Enterprises	Logistics Output	Logistics Employment	Average Wage of Employed Persons
**Weighting**	0.11	0.08	0.07	0.07
**Indicators**	**Investment in fixed assets**	**Cargo volume**	**Freight transport turnover**	**Post and telecommunications business volume**
**Weighting**	0.09	0.08	0.14	0.10
**Indicators**	**Mobile phone penetration rate**	**Average output of employed persons**	**Growth rate of output**	**Growth rate of output**
**Weighting**	0.08	0.06	0.07	0.05

**Table 3 ijerph-19-15082-t003:** Indicator weights for ecology.

Indicators	SO_2_ Emissions	Number of Environmentally Sound Treatment Plants	Growth Rate of Investment in Ecological Restoration
**Weighting**	0.07	0.11	0.26
**Indicators**	**Amount of domestic waste removed**	**Nitrogen oxide emissions**	**Total water resources**
**Weighting**	0.12	0.04	0.16
**Indicators**	**Growth rate of investment in industrial pollution control**	**Growth rate of afforested area**	**Particulate emissions**
**Weighting**	0.07	0.13	0.04

**Table 4 ijerph-19-15082-t004:** Classification of coupling coordination types.

Coordinate Numerical	Coordination Degree	Coordination Level	$Z_{a}^{n}>Z_{b}^{n}\;$	$Z_{a}^{n}<Z_{b}^{n}\;$
0.0–0.1	Imbalance type	Extreme imbalance	The level of ecological environment development is lagging behind	The development level of regional logistics lags behind
0.1–0.2		Serious imbalance	The level of ecological environment development is lagging behind	The development level of regional logistics lags behind
0.2–0.3		Moderate imbalance	The level of ecological environment development is lagging behind	The development level of regional logistics lags behind
0.3–0.4		Mild imbalance	The level of ecological environment development is lagging behind	The development level of regional logistics lags behind
0.4–0.5	Transitional type	On the verge of imbalance	The level of ecological environment development is lagging behind	The development level of regional logistics lags behind
0.5–0.6		Barely coordinated	The level of ecological environment development is lagging behind	The development level of regional logistics lags behind
0.6–0.7	Coordinate type	Primary coordination	The level of ecological environment development is lagging behind	The development level of regional logistics lags behind
0.7–0.8		Intermediate coordinate	The level of ecological environment development is lagging behind	The development level of regional logistics lags behind
0.8–0.9		Good coordination	The level of ecological environment development is lagging behind	The development level of regional logistics lags behind
0.9–1.0		Vintage coordination	The level of ecological environment development is lagging behind	The development level of regional logistics lags behind

**Table 5 ijerph-19-15082-t005:** Coupling coordination degree of regional logistics and ecological environment from 2010 to 2019.

	2010	2011	2012	2013	2014	2015	2016	2017	2018	2019
Beijing	0.62	0.60	0.66	0.59	0.59	0.55	0.63	0.63	0.58	0.52
Tianjin	0.50	0.63	0.54	0.56	0.63	0.52	0.51	0.58	0.45	0.52
Hebei	0.60	0.59	0.61	0.54	0.69	0.62	0.66	0.56	0.66	0.51
Shanxi	0.48	0.54	0.50	0.47	0.54	0.51	0.51	0.52	0.51	0.42
Inner Mongolia	0.54	0.57	0.57	0.55	0.54	0.54	0.56	0.49	0.52	0.45
Liaoning	0.65	0.59	0.67	0.59	0.59	0.58	0.50	0.54	0.57	0.45
Jilin	0.50	0.49	0.49	0.49	0.52	0.54	0.51	0.49	0.48	0.41
Heilongjiang	0.51	0.55	0.50	0.49	0.52	0.56	0.52	0.49	0.46	0.44
Shanghai	0.67	0.64	0.68	0.58	0.71	0.70	0.71	0.58	0.67	0.61
Jiangsu	0.65	0.72	0.68	0.68	0.71	0.71	0.68	0.67	0.70	0.62
Zhejiang	0.72	0.71	0.73	0.67	0.70	0.73	0.72	0.67	0.69	0.66
Anhui	0.58	0.56	0.60	0.71	0.60	0.62	0.61	0.58	0.64	0.54
Fujian	0.63	0.66	0.62	0.57	0.63	0.72	0.64	0.59	0.65	0.58
Jiangxi	0.55	0.54	0.56	0.54	0.54	0.58	0.61	0.53	0.54	0.50
Shandong	0.72	0.67	0.71	0.60	0.65	0.63	0.66	0.66	0.67	0.62
Henan	0.55	0.61	0.58	0.57	0.61	0.63	0.61	0.60	0.60	0.56
Hubei	0.63	0.57	0.59	0.59	0.64	0.65	0.65	0.63	0.63	0.58
Hunan	0.63	0.58	0.65	0.55	0.63	0.63	0.60	0.61	0.60	0.55
Guangdong	0.85	0.77	0.82	0.84	0.80	0.87	0.82	0.81	0.83	0.76
Guangxi	0.55	0.59	0.55	0.54	0.58	0.61	0.59	0.57	0.59	0.52
Hainan	0.49	0.47	0.51	0.43	0.46	0.50	0.48	0.49	0.43	0.50
Chongqing	0.54	0.54	0.51	0.52	0.50	0.57	0.53	0.55	0.57	0.49
Sichuan	0.61	0.65	0.62	0.63	0.70	0.67	0.69	0.64	0.63	0.59
Guizhou	0.47	0.52	0.48	0.50	0.53	0.58	0.58	0.54	0.54	0.48
Yunnan	0.52	0.52	0.52	0.48	0.53	0.56	0.56	0.55	0.63	0.52
Shanxii	0.54	0.53	0.53	0.50	0.56	0.53	0.57	0.53	0.57	0.49
Gansu	0.43	0.47	0.46	0.38	0.49	0.48	0.49	0.45	0.50	0.38
Qinghai	0.45	0.47	0.43	0.42	0.43	0.49	0.48	0.46	0.49	0.41
Ningxia	0.42	0.50	0.44	0.40	0.45	0.44	0.50	0.41	0.43	0.36
Xinjiang	0.53	0.49	0.55	0.49	0.53	0.57	0.52	0.52	0.50	0.47
Average	0.57	0.58	0.58	0.55	0.59	0.60	0.59	0.57	0.58	0.52
Maximum	0.85	0.77	0.82	0.84	0.80	0.87	0.82	0.81	0.83	0.76
Minimum	0.42	0.47	0.43	0.38	0.43	0.44	0.48	0.41	0.43	0.36

**Table 6 ijerph-19-15082-t006:** Level of coupling coordination between logistics and ecological environment in seven regions in China.

	2010	2011	2012	2013	2014	2015	2016	2017	2018	2019
North China	0.55	0.59	0.59	0.56	0.61	0.55	0.58	0.56	0.55	0.50
Northeast China	0.56	0.56	0.56	0.54	0.56	0.58	0.52	0.51	0.51	0.44
East China	0.65	0.65	0.66	0.63	0.66	0.68	0.67	0.62	0.66	0.60
Central China	0.61	0.59	0.61	0.57	0.63	0.64	0.62	0.61	0.61	0.56
South China	0.66	0.63	0.66	0.63	0.64	0.69	0.66	0.65	0.65	0.63
Southwest region	0.54	0.56	0.54	0.54	0.57	0.60	0.59	0.57	0.59	0.52
Northwest region	0.48	0.50	0.49	0.44	0.50	0.51	0.51	0.48	0.50	0.43

**Table 7 ijerph-19-15082-t007:** Influencing factors and variable explanations of regional logistics and ecological environment development level.

	Variable	Interpretation of Meaning	Unit
Explained variable	Coupling coordination level (*Y*)	Degree of coupling coordination	
Explanatory variables	Level of technological development (*X*_1_)	Number of patents granted in the year	Piece
	Level of economic development (*X*_2_)	GDP per capita	CNY ten thousand
	Industrial pollution control (*X*_3_)	Industrial governance investment/gross regional product	
	The industrial structure (*X*_4_)	The proportion of industrial industry in GDP	%
	Level of transport facilities (*X*_5_)	(railway mileage + highway mileage)/population	Km/person
	Waste management degree (*X*_6_)	Number of harmless treatment plants	Unit

**Table 8 ijerph-19-15082-t008:** Descriptive statistics of variables.

Variable	Obs	Mean	Std. Dev.	Min	Max
*Y*	300	0.57	0.09	0.36	0.87
*X* _1_	300	4.93	7.35	0.03	52.74
*X* _2_	300	5.19	2.65	1.03	16.42
*X* _3_	300	12.17	11.04	0.67	99.19
*X* _4_	300	0.38	0.09	0.11	0.63
*X* _5_	300	38.90	24.01	5.35	141.79
*X* _6_	300	28.96	18.50	3.00	111.00

**Table 9 ijerph-19-15082-t009:** Moran index values for each indicator.

	Regional Logistics Composite Index		Comprehensive Index of Ecological Environment		Coupling Coordination Composite Index	
	*Moran’s I*	*p*	*Moran’s I*	*p*	*Moran’s I*	*p*
2010	0.25 ***	0.01	0.32 ***	0.00	0.18 **	0.03
2011	0.23 ***	0.01	0.12	0.09	0.17 **	0.03
2012	0.21 ***	0.01	0.20 ***	0.02	0.20 ***	0.02
2013	0.19 ***	0.02	0.11	0.09	0.17 **	0.03
2014	0.17 **	0.03	−0.02	0.44	0.11 **	0.09
2015	0.23 ***	0.01	0.51 ***	0.00	0.27 ***	0.00
2016	0.16 **	0.04	0.24 ***	0.01	0.17 ***	0.03
2017	0.22 ***	0.01	0.19 ***	0.02	0.22 ***	0.01
2018	0.18 ***	0.02	0.26 ***	0.00	0.20 ***	0.02
2019	0.22 ***	0.01	0.38 ***	0.00	0.34 ***	0.00

Note: **, and *** represent significance at the 5%, and 1% levels, respectively.

**Table 10 ijerph-19-15082-t010:** Selection and test of spatial econometric model.

Test Methods		Test Results
LM	LM Lag	23.01 ***
	LM Error	39.47 ***
	R-LM Lag	1.19
	R-LM Error	17.65 ***
Wald	Wald spatial lag	13.78 **
	Wald spatial error	13.35 **
LR	LR spatial lag	67.97 ***
	LR spatial error	67.00 ***
Hausmann		0.00 ***

Note: **, and *** represent significance at the 5%, and 1% levels, respectively.

**Table 11 ijerph-19-15082-t011:** SDM model estimation results under different models.

	Time Fixed Model		Spatial Fixed Model		Spatiotemporal Dual Fixation Model	
	Coefficient	*p*	Coefficient	*p*	Coefficient	*p*
*X* _1_	1.56	0.12	−2.03 **	0.04	1.76 *	0.08
*X* _2_	4.09 ***	0.00	−0.93	0.36	−0.71	0.48
*X* _3_	−3.82 ***	0.00	0.5	0.62	−0.33	0.74
*X* _4_	4.62 ***	0.00	1.71 *	0.07	1.90 *	0.06
*X* _5_	−4.25 ***	0.00	0.51	0.61	1.08	0.28
*X* _6_	8.71 ***	0.00	1.58	0.11	1.86 *	0.06
W∗*X*_1_	6.02 ***	0.00	2.59 **	0.01	2.87 ***	0.04
W∗*X*_2_	−3.30 ***	0.01	1.95 *	0.05	2.10 **	0.04
W∗*X*_3_	−3.49 ***	0.00	−0.62	0.54	−1.89 *	0.06
W∗*X*_4_	−0.67	0.5	−1.4	0.16	0.11	0.91
W∗*X*_5_	−2.00 **	0.05	−0.16	0.87	0.71	0.48
W∗*X*_6_	−3.42 ***	0.01	−3.61 ***	0.00	−3.07 ***	0.02
rho	−1.5	0.13	8.63 ***	0.00	1.20 *	0.09
sigma2_e	12.14 ***	0.00	12.04 ***	0.00	12.25 ***	0.00

Note: *, **, and *** represent significance at the 10%, 5%, and 1% levels, respectively.

**Table 12 ijerph-19-15082-t012:** Three effect estimation results.

	(1)	(2)	(3)
Variable	Direct Effect	Indirect Effect	Total Effect
*X* _1_	1.31 (−0.19)	5.60 *** (0.00)	6.30 *** (0.00)
*X* _2_	4.30 *** (0.00)	−3.69 *** (0.00)	−1.52 (−0.13)
*X* _3_	−3.52 *** (0.00)	−3.33 *** (0.00)	−5.37 *** (0.00)
*X* _4_	4.87 *** (0.00)	−0.92 (−0.36)	1.18 (−0.24)
*X* _5_	−4.30 *** (0.00)	−1.94 * (−0.05)	−4.36 *** (0.00)
*X* _6_	9.29 *** (0.00)	−3.83 *** (0.00)	−0.07 (−0.95)
Observations	300.00	300.00	300.00
R-squared	0.68	0.68	0.68
Number of code	30.00	30.00	30.00

Note: *, and *** represent significance at the 10%, and 1% levels, respectively.

**Table 13 ijerph-19-15082-t013:** Model estimation results under different weight matrices.

	Inverse Geographic Distance Square Matrix		Adjacent Weight Matrix	
	Coefficient	*p*	Coefficient	*p*
*X* _1_	−1.32	0.19	−1.76 *	0.08
*X* _2_	−0.62	0.54	−0.71	0.48
*X* _3_	−0.16	0.87	−0.33	0.74
*X* _4_	2.11 **	0.03	1.90 *	0.06
*X* _5_	0.58	0.56	1.08	0.28
*X* _6_	2.30 **	0.02	1.86 *	0.06
W∗*X*_1_	2.34 **	0.02	2.87 ***	0.04
W∗*X*_2_	0.11	0.91	2.10 **	0.04
W∗*X*_3_	−2.46 **	0.01	−1.89 *	0.05
W∗*X*_4_	0.33	0.74	0.11	0.91
W∗*X*_5_	1.95	0.05	0.71	0.48
W∗*X*_6_	−0.68	0.49	−3.07 ***	0.02
rho	1.88 **	0.04	1.20 *	0.09
sigma2_e	12.24 ***	0.00	12.25 ***	0.00

Note: *, **, and *** -represent significance at the 10%, 5%, and 1% levels, respectively.

## Data Availability

All the original data used in this paper are from the dataset of *China Urban Statistical Yearbook* from 2010 to 2019. The source website is as follows: https://data.stats.gov.cn/index.htm (accessed on 3 January 2021).

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
