# Peer review of "Spatial Econometric Analysis of the Level and Influencing Factors of Coupling and Coordination between Regional Logistics and the Ecological Environment in China"

_ijerph, 2022, doi:10.3390/ijerph192215082_

Round 1

Reviewer 1 Report

1.       The title of the article is not sufficiently professional, and "Research on ......" is not recommended.

2.       Inadequate summary of article title, "Regional Logistics", conducting a case study?

3.       The Abstract section does not adequately summarize the study, and the concluding remarks are general in nature and not unique or highly instructive.

4.       The discussion of the mechanism of action between logistics and ecological environment is not clear. The article Fig1 and Fig2 show the mechanism of action of logistics activities on ecological environment and the mechanism of action of ecological environment on logistics activities, respectively, and the lack of text outlining the mechanism of action of both is not sufficient and slightly thin.

5.       The literature on regional logistics and ecological environment is insufficiently researched and lacks logic. The article carries out literature research in 3 aspects: the impact of regional logistics on the environment, the coordination of economic development and ecological environment, and the coupled coordination theory. The literature [13] shows that the correlation and coordination between economic development and ecological environment are weakly correlated, and the literature [14] and [15] show that the coordination needs to be strengthened, is this part to introduce the "coordination model"? The article then discusses the close relationship between logistics development and the ecological environment, which lacks a supporting basis and is abruptly connected. In addition, why is the close correlation between regional logistics and economic development? The discussion does not lead to the theme of the study adequately and naturally, and the research is not well argued and lacks organization.

6.       What is the innovation that sets the article apart from other studies needs to be specifically described in the Introduction.

7.       The discussion of the research methodology is incomplete. Line 136, "This paper will introduce a spatial econometric model ......", in context, lacks an explanation of the spatial econometric model and its spatial effects Relevant explanation.

8.       2.1. Construction of index system section, the selection of indicators lacks objectivity, part of the content redundant, the main priority is not clearly distinguished. Line 147 namely...... to 157 content and delete, why the secondary indicators of regional logistics level choose these 12, lack of theoretical basis, why the secondary indicators of environmental development level choose 9, lack of theoretical support.

9.       Article selected data 2010-2019, sample data can be updated to 2020 or 2021, why did you choose provincial data? The provincial data are macro in nature, which provinces were excluded from the study? Excluding relevant provinces simply because of a lack of data is not a sufficient explanation, why not choose prefecture-level city data?

10.    2.3. the Measurement of the level of coupling coordination section, the relevant delineation data for the measurement of the level of coupling coordination lacks a theoretical support basis.

11.    In line 294, what is the basis for the explanatory and accepted variables selected for the spatial effects analysis and how do the six explanatory variables reflect the regional logistics system?

12.    The spatial effects are biased towards the analysis of the correlation between the level of economic development and the ecological environment, and the discussion is a bit off the mark.

13.    The article could conclude by adding insights into future research trends.

14.    Considering the cutting-edge nature of the literature, the literature could be further referenced to the latest research.

Reviewer 2 Report

I have gone through the article "Research on the Level of Coupling and Coordination Relation-2 ship between Regional Logistics and Ecological Environment 3 in China". This is an interesting topic that is closely related to the scope of IJERPH. However, the manuscript should be improved in the following areas.

Abstract

(0)  Please includes the basic design of the study.

1. Introduction

(1) Please improve the introduction to be clearly stated the difference between this study and the existing study.

(2) The authors must still emphasize the originality and the motivations of the study.

(3) This section is necessary for you to clarify the "contribution" of your study.

(4) Please improve the introduction to be clearly stated research questions.

2. Research Methods

(5) Please give the basis for establishing the indicator system (including references, models, etc.) in order to convince the reader.

5. Conclusions

(6) Please include thoughts on future research in the conclusion section.

Reviewer 3 Report

Dear Authors,

thank you for the interesting and valuable paper.

My comments and questions are of an improving nature.

1) in the introduction section you refer, however not directly, to reducing the knowledge gap. Are there any utiltarian aspects of your research?

2) In the methodology description you do not refer to any constraints,  geographical constraints are mentioned in the conclusion section. Are there any other constraints that should be considered? How about the data? It comes from different sources, was it collected with the same methodology to ensure the scientific integrity of the result? Or it does not matter?

The paper is well-structured, and clearly written, and the conclusion section covers many aspects. The results can be used as a benchmark or can be compared with others.

Regards

Round 2

Reviewer 1 Report

1.English language and style are fine/minor spell check required.

2.Index system should be expounded more detail.

Author Response

Response to Reviewer1

Dear Reviewer:

      I’m very appreciate for your valuable advice. I have studied the valuable comments from you which give me a great deal of inspiration. The point to point responds to your comments are listed as following:

Point 1: English language and style are fine/minor spell check required.

Response 1:

Thank you for your suggestion. English language and style of this article has been completely revised. It can be seen in the revised manuscript.

Point 2: Index system should be expounded more detail.

Response 2:

Thank you for your careful comment. The modification results are as follows.

Among them, the dimension of development scale represents the size of the regional logistics industry development, which is mainly evaluated from the quantitative aspects, including the number of enterprises, output value, business volume and other quantitative performance indicators. The dimension of development level focuses on the development quality, including four indicators, namely, the number of employees in the logistics industry, the average wage of employees, the average output value, and fixed asset investment. The dimension of informatization level selects mobile phone penetration rate as the measurement indicator to analyze and control logistics information. The dimension of development potential chooses the incremental ratio of industrial development and employment growth for measurement. 

Among them, the dimension of environmental pollution selects carbon dioxide emissions, nitrogen oxide emissions, and particulate emissions as environmental pollution indicators. The dimension of environmental remediation selects the domestic waste clearing and transportation volume and the number of harmless treatment plants as the measurement indicators. In the dimension of natural resources, water resources, as the core resource of the ecosystem, is a necessary condition to ensure the ecological balance and development, so the indicator of total water resources is selected. Finally, in the dimension of development potential, this paper adopts the relative incremental measurement index, and selects the growth rate of industrial pollution control, the growth rate of afforestation area, and the growth rate of ecological restoration investment.
